# The Impact of Having One Parent Absent on Children’ Food Consumption and Nutrition in China

**DOI:** 10.3390/nu11123077

**Published:** 2019-12-17

**Authors:** Xu Tian, Hui Wang

**Affiliations:** 1China Center for Food Security Studies & College of Economics and Management, Nanjing Agricultural University, Nanjing 210095, China; xutian@njau.edu.cn; 2Department of Epidemiology and Biostatistics, School of Public Health, Nanjing Medical University, Nanjing 210095, China

**Keywords:** single-parent children, food consumption, nutrition intake, propensity score matching, compensation effect

## Abstract

The rapid economic and social development in the past decades has greatly increased the societal acceptance of divorce and non-marital pregnancies in China, which leads to a soaring number of single-parent children. This paper aimed to investigate the impact of having one parent absent on children’ food consumption and nutrition status. We extracted 1114 children from a longitudinal household survey data in China, all of which were observed twice. Using the Propensity Score Matching and Difference-in-Difference methods, we found that being raised by one parent does not have a negative effect on children’s food consumption and nutrition intake. On the contrary, single-parent families tend to provide more food to their children as a compensation for the absence of one parent and this compensation effect offsets the negative impact caused by declined family income. Particularly, urban, rich families had stronger compensation effect than other families with low and middle incomes.

## 1. Introduction

For thousands of years, traditional marriage values were passed down from generation to generation in China, where children were very unlikely to live with a single mother. However, the women’s emancipation movement inspired by the May Fourth Movement in 1919 and the egalitarian movement launched during chairman Mao’s time have challenged the tradition of son preference and increased the status of women in the family [1,2]. The transition from a traditional to modern society was further sped up after late 1970s due to the implementation of the family planning policy and the reform and opening up policy. Associated with the rapid economic and social development, Chinese people’s attitudes toward marriage and family values have changed significantly. In particular, the societal acceptance of divorce and non-marital pregnancies has increased significantly. As a result of rising divorce rates, especially in the early years of marriage, the number of single-parent children has soared [3,4]. The Ministry of Civil Affairs (MCA) in China claimed that the crude divorce rate increased dramatically from 0.09% in 2002 to 0.3% in 2016, and 4.16 million couples divorced in 2016 [5]. The soaring divorce rate led to a raising prevalence of single-parent families. A report released by the National Health and Family Planning Commission (NHFC) announced that single-parent families increased year by year, reached 23.96 million in 2010, and were mainly caused by divorce [6].

The well-being of single-parent children has drawn attention from many researchers in China. The research mainly focuses on psychological health, educational performance, and legal aid [7,8,9,10], while a few researches have investigated their dietary quality such as food consumption and nutrition status [11]. Dietary quality is vital for children’ growth and human capital accumulation. Studies already found significant difference in food structure between single-parent children and dual-parent children in Jamaica [12], America [13,14], and Korea [15].

However, single-parent children differ from dual-parent children in many various aspects such as family income, population structure, and religious belief, all of which might have impact on children’ diets. Previous literature also found a significant difference in occupation of parents [14], household income [13,16,17], and family structure [12] between single-parent families and dual-parent families. Therefore, direct comparison of food consumption between single-parent children and dual-parent children might be biased due to counter factors like heterogeneity in family characteristics. Thus, we need a more precise evaluation of the impact of one parent’s absence on children’ food consumption and nutrition status.

In addition, single-parent children are usually more vulnerable compared to their counterparts living in dual-parent families due to the deteriorated family income [18]. However, parents and grandparents of single-parent children might compensate children by spending more time with them. Previous studies also found that single mothers/fathers in England spend more time taking care of their children compared with parents from dual-parent families [19]. Similar evidence was also detected in the US [20]. Moreover, previous literature already showed that China was undergoing a rapid nutrition transition in the past decades. The traditional Chinese diet, which is high in complex carbohydrates and fiber, had been gradually replaced by a refined food and Western food diet, which is high in fat, saturated fat, and sugar [11,21,22,23]. As a result, the food accessibility and dietary diversity had been significantly improved in China. Given the fact that parents (and grandparents) in China tend to reward children by giving more food like snacks for their good behavior [24], we suspect that single parents in China might also compensate their children by offering more food, which might result in a positive impact on single-parent children’ dietary quality.

Therefore, this study aimed to first compare the consumption of 7 representative food items and the intake of 12 nutrients between single-parent children and their counterparts from dual-parent families. Using a Propensity Score Matching and Difference-in-Difference method (PSM-DID), we estimated the average treatment effect of the treated (ATT) for single-parent children, which was the pure difference between single-parent children and dual-parent children caused by having one parent absent. Moreover, we also decomposed the pure difference into income and compensation effects. The former referred to the negative effect caused by reduction in family income and the latter referred to the food compensation provided by other family members, both of which were the main affecting channels on children’s diet after one parent left the family. This was different from previous studies which simply compared food consumption and nutrition between children from different families. We contributed to the literature by providing a more precise comparison using longitudinal dataset and matched samples. In addition, after decomposing the total effect into income effect and compensation effect, we provided further evidence for a possible mechanism to explain the differences between single-parent and dual-parent children.

## 2. Materials and Methods

### 2.1. Data

The data used in this study was drawn from the recent four waves (2004, 2006, 2009, and 2011) of the China Health and Nutrition Survey (CHNS). The CHNS was a national representative longitudinal survey jointly conducted by the Carolina Population Center at the University of North Carolina at Chapel Hill (UNC-CH) and by the National Institute for Nutrition and Health at the Chinese Center for Disease Control and Prevention (CCDC). More than 4000 households each year were surveyed through a multi-stage, random cluster survey in 9 provinces (Guangxi, Guizhou, Henan, Heilongjiang, Hubei, Hunan, Jiangsu, Liaoning, and Shandong; three municipalities, Beijing, Chongqing, and Shanghai, were included in 2011). The survey collected abundant information on food consumption and socio-economic characteristics for each individual and family. This survey was approved by the institutional review boards of the aforementioned institutions. All participants provided written informed consent. Detailed information about the data can be found elsewhere [25].

### 2.2. Sample

Only children below 18 years old were included in our sample. Single-parent children were defined as those whose mother or father do not live in the family with them due to death, divorce, separation, or abandonment [26]. Here, our definition is based on foster relationship rather than kinship. Children were defined as dual-parent children if they lived together with their mother and father, including biological parents, stepparents, and foster parents. Furthermore, single-parent children were classified into single-mother children (those who live with single mother) and single-father children (those who live with single father). We selected two period surveys for each child. Those who lived with both parents in all two periods were treated as control group (dual-parent children), and those who lived with both parents in the first period while only live with one parent in the second wave were defined as treated group (single-parent children). Orphans were censored because they were raised in a very different environment. Finally, after matching data from various individual characteristics and socio-economic conditions of family, we got 1114 children (2228 observations), including 974 dual-parent children, 97 single-mother children, and 43 single-father children (Selection of sample is presented in Appendix A in the Appendix A).

In the robustness check section, single-parent children were divided into two subgroups: single-parent children living in urban, rich families (per capita income greater than the average income in urban area) and single-parent children living in rural, poor families (per capita income smaller than the average income in rural area). The sample sizes for these two subgroups were 94 and 16 children, respectively.

### 2.3. Measurement of Variables

In order to measure the food consumption and nutrition status of children, we adopted 7 indices to measure dietary structure and 12 indices to measure nutrition intake. The 7 food indices were the daily consumption of cereal grains, vegetables, meat (including poultry), aquatic products, eggs, dairy products, and fruits. The 12 nutrition indicators were the daily intake of calories; carbohydrates; fats; proteins; vitamins A, B, C, and E; calcium; iron; zinc; and selenium. Food consumption was recorded using a 24-hour recalling method, and nutrient intake was calculated by converting food consumption using the China Food Composition Table [27].

### 2.4. Empirical Model

To test whether single-parent children have poor dietary quality compared to dual-parent children, we adopted the PSM-DID method to estimate the ATT of food consumption and nutrition intake for single-parent children, which can be taken as the pure effect caused by being raised by one parent. The PSM attempted to mimic randomization by creating two samples that are comparable on all observed covariates, which was commonly used to adjust for the bias from selection into the treatment group (experimental group) conditional on observed variables [28,29], and the DID method was widely adopted to deal with selection on time-invariant unobservable factors. The PSM-DID was achieved by using PSM with the baseline data to make certain that the comparison group was similar to the treatment group before treatment and then by applying DID to the matched sample so that the observable heterogeneity in the initial conditions could be dealt with and so that the control group and the treatment group could be compared directly [30,31]. In this study, we adopted the kernel matching method and the Epanechnikov kernel density function. The difference between the control group and the treatment group estimated using PSM-DID can be specified as follows:
(1)$$PSM-DID=E\left[N_{1i}^{T}-N_{0i}^{T}|P\left(X_{0i}\right),T=1\right]-E\left[N_{1i}^{C}-N_{0i}^{C}|P\left(X_{0i}\right),T=0\right]\;$$
where $N_{1i}^{T}$ and $N_{0i}^{T}$ are the outcomes of treatment individual i (single-parent children) before intervention and after intervention respectively. $N_{1i}^{C}$
$N_{0i}^{C}$ is the pre- and post-intervention difference in outcome of control individual i (dual-parent children). T is treatment dummy, which refers to treatment group when it equals 1. In the initial period, all children live with both parents; while in the following-up period, treatment group lose one of their parents (single-parent children), and control group still stay with both parents $P\left(X_{0i}\right)$ is the propensity score estimated using matching variables in the initial period, which includes characteristics of children (age and gender), family (income per capita, household size, and share of children in the family), and household head (age, physical activity, gender, and education). In addition, the time gap between two periods and regional dummies were also controlled in the model. 

To decompose the pure difference between single-parent children and dual-parent children (ATT) into income effect and compensation effect, we first estimated the income elasticity of various food items and nutrient intake. Income elasticity of food consumption was estimated using a linear-log function (see Equation (2)), and income elasticity of nutrient was estimated using a direct method suggested in previous studies [32,33].
(2)$$N_{it}=\beta_{0}+\beta_{1}lnincome_{it}+\beta_{2}father_{i}*lnincome_{it}+\beta_{3}mother_{i}*lnincome_{it}+\sum_{j=1}^{p}\lambda_{j}Z_{itj}+v_{i}+\varepsilon_{ij}$$
where $N_{it}$ refers to various food and nutrition indicators; $father_{i}$ and $mother_{i}$ are dummy variables for single-father and single-mother children. $lnincome_{it}$ is the logarithm of real family net income per capita deflated to 2004, which is the sum of all sources of income and revenue minus expenditures such as income from business, farming, fishing, gardening, livestock, non-retirement wages, retirement income, subsidies, and other income. $Z_{itj}$ refers other co-variants that may also affect children’s food consumption and nutrition status such as characteristics of children (age) and household head (age, physical activity, and education), as suggested in previous studies [21,22]; and $v_{i}$ refers to time-invariant individual effects. 

To test the robustness of the results, we also adopt a fixed elasticity model to estimate income elasticity as follows:
(3)$$\ln N_{it}=\beta_{0}+\beta_{1}lnincome_{it}+\beta_{2}father_{i}*lnincome_{it}+\beta_{3}mother_{i}*lnincome_{it}+\sum_{j=1}^{p}\lambda_{j}Z_{itj}+v_{i}+\varepsilon_{ij}$$

In the second step, income effect was calculated by multiplying the income elasticity with income change between two periods. Income change during the two periods was measured using the DID method, where characteristics of household (household size and ratio of children), household head (age, occupation, gender, and education), regional dummy variables (urban dummy and provincial dummies), and time gap between two periods were selected as co-variants. Compensation effect was estimated by subtracting the income effect from the pure effect (ATT).

The equality test of mean values between single-parent children and dual-parent children was conducted by ANOVA, which accounts for unequal sample sizes between different groups. All empirical analyses were conducted using Stata 14.0 SE (StataCorp, Texas, USA).

This survey was approved by the institutional review boards of the University of North Carolina at Chapel Hill (UNC-CH) and of the Chinese Institute of Nutrition and Food Safety (INFS) at the Chinese Center for Disease Control and Prevention (CCDC). All participants provided written informed consent, and all methods were performed in accordance with the relevant guidelines and regulations. In addition, the ethics committees of the Medical Faculty of the University of Goettingen and of the University of North Carolina at Chapel Hill approved our use of this data in 2013 (Application no. 26/6/13 An).

## 3. Results

### 3.1. Descriptive Analysis of Variables

Descriptive analysis of those co-variants is presented in Table 1. We find significant difference between children from different types of families.

### 3.2. Comparison before Matching

Table 2 presents a comparison of representative food consumption and nutrition intake between single-parent children and dual-parent children before matching. Single-parent children had lower consumption of all food items except for vegetables. They also had lower intake of all nutrients except for carbohydrates. Similar results were detected after dividing single-parent children into single-mother children and single-father children. In particular, single-mother children had significantly lower consumption of aquatic and dairy products (all *p* < 0.05) and their fat intake was significantly lower than that of dual-parent children (*p* < 0.01). Differently, single-father children had significantly lower consumption of most food items except for vegetables and dairy products than dual-parent children (all *p* < 0.05) and they had lower intake of all nutrients, of which 8 (calorie, fat, protein, vitamins B and E, iron, zinc, and selenium) were statistically significant (all *p* < 0.05).

### 3.3. ATT of Food Consumption and Nutrition Intake

Results shown in Table 2 might be biased due to pre-treatment heterogeneity. We thus adopted the PSM-DID to estimate the ATT, which could be treated as the “pure effect” caused by being raised by one parent. Only results of ATT are presented in Table 3, and the balance test and mean comparison using matched sample are presented in the Appendix A (Appendix A). In general, we did not observe strong evidence that being raised by one parent had a significant impact on food consumption and nutrition intake of single-parent children. By contrast, ATT of most indicators was positive, indicating that the positive compensation effect might be greater than the negative income effect. Particularly, single-mother children had significantly higher consumption of eggs and fruits than dual-parent children (all *p* < 0.05), but the differences in nutrition intake between them were insignificant. Differently, single-father children had significantly higher consumption of meat and aquatic products but lower consumption of cereals (all *p* < 0.05), which resulted in a significantly higher intake of fat and vitamin C (all *p* < 0.05). 

### 3.4. Income Effect and Compensation Effect

The estimated ATT could be taken as the total effect caused by the absence of one parent after matching. We further decomposed it into income effect and compensation effect as mentioned earlier.

To calculate income effect, we first estimated the income elasticity of various food items and nutrients and presented the results in the Appendix A (Appendix A). Incomes at each period and income change caused by having one parent absent (ATT of income) are presented in the Appendix A as well (Appendix A). We found that income growth in all families during these two periods and household income of dual-parent families increased by 43.62%, which was much higher than the income growth in single-parent families. As a result, the ATTs of income for all types of single-parent families were negative, which led to a negative shock on the dietary quality of single-parent families.

The income effect was estimated by multiplying the ATT of income with income elasticity, and the compensation effect was measured by the difference between the total pure impact (ATT) and income effect (Table 4). In general, income effect was very small in almost all dietary indicators and compensation effect dominated the total effect. Particularly, single-mother children were compensated by fruits, eggs, and aquatic products and single-father children were compensated by vegetables, meat and poultry, aquatic products, fruit, and eggs. By contrast, negative compensation of cereals and dairy products was observed for both single-mother and single-father children. Food compensation resulted in an improvement in nutrition status. Results showed that single-mother children had a higher intake of total calories, carbohydrates, vitamins C and E, zinc, and selenium; on the other hand, single-father children were detected to have higher intake of fat, vitamin C, and calcium.

### 3.5. Robustness Check

To test the robustness of the results, we first employed three different kernel density functions (biweight, uniform, and tricube) to estimate ATT (Appendix A). The main results were consistent with the current one, indicating our findings are robust.

We also reestimated the income effect by using a fixed income elasticity function as presented in Equation (3). Results were presented in the Appendix A (Appendix A). Similar to the results presented in Table 4, we found that income effect was negligible and that compensation effect dominated the total effect.

In addition, there was a substantial difference between rural and urban areas in terms of cuisine (Appendix A) and food availability [23], which might lead to different ways of compensating single-parent children. We thus compared the food consumption and nutrition intake between single-parent children living in urban, rich families and dual-parent children living in urban areas as well as single-parent children living in rural, poor families and dual-parent children living in rural areas. The estimated ATT is presented in Table 5. We found significant positive ATT of vegetables and fruits in urban single-parent families (all *p* < 0.05) and significant positive ATT of cereals in rural single-parent families (*p* < 0.05). In addition, single-parent children living in urban, rich families had a significantly higher intake of most nutrients such as calories; carbohydrates; fats; vitamins B, C, and E; calcium; and selenium but lower intake of vitamin A (all *p* < 0.05). By contrast, single-parent children living in rural, poor families only had significantly higher intake of vitamin A (*p* < 0.05). Similar comparison was also conducted for single-parent children living in urban, poor families and dual-parent children living in urban areas, as well as single-parent children living in rural, rich families and dual-parent children living in rural areas. Results were presented in Appendix A. Positive compensation was still detected in urban, poor families but not as strong as that in urban, rich families. On the contrary, negative ATT was found in rural, poor families for dairy products; calorie; and all three macronutrients iron, zinc, and selenium (all *p* < 0.1).

## 4. Discussion

This paper adopted the recent four waves of CHNS data to explore the shock of losing one parent on children’ food consumption and nutrition intake. We constructed a two-period balanced panel data and employed the PSM-DID method to estimate the ATT on food consumption and nutrition intake of single-parent children. Results showed that single-parent children had a poorer dietary status compared with dual-parent children before matching. However, this difference was mainly caused by the pretreatment heterogeneity between two types of families. Once we removed these heterogeneities that might affect children’ food consumption and nutrition intake and matched single-parent families with dual-parent families that had similar characteristics, single-parent children had even better dietary status than their counterparts living in dual-parent families. In particular, absence of a father would increase the consumption of eggs and absence of a mother would increase the consumption of meat and poultry and of aquatic products of single-father children but the consumption of cereals by single-father children decreased. In addition, no significant negative impact was observed on nutrition status of single-mother children but a significant positive impact was detected on the intake of fat and some vitamin for single-father children. Those results are robust to using various kernel density functions. To explore the potential mechanism behind these findings, we decomposed the total impact into a negative impact caused by losing one parent and a positive impact from families’ compensation. We found that income change only had a very small impact on children’ dietary quality and that the compensation effect dominated the total impact and led to a positive total impact on some dietary indicators.

The small income effect was mainly caused by the small income elasticity, so that income change would not have a strong impact on children’ food consumption, which was consistent with findings in previous studies [33,34,35]. Therefore, even though the impact of having one parent absent had a strong negative impact on family income, the income effect was still quite small due to negligible income elasticity. In addition, overconsumption of food could make people feel uncomfortable and could lead to health problems; thus, income increase might lead to substitution of low-quality food by high-quality food rather than higher consumption of each food items.

We found a strong compensation effect in terms of food consumption. Previous literature already found that single-parent tended to spend more time taking care of children compared with dual-parent in the UK [19] and the US [20], and it was believed to be the compensation to single-parent children provided by a single parent. We provided new evidence to the literature that single-parent families in China might provide more food to single-parent children as a compensation for one parent absent. Previous literature also found that girls from single-parent families had a significantly greater daily intake of energy and fat than girls from dual-parent families in England [36].

The different compensations between single-mother children and single-father children could be attributed to the heterogeneous preference of single-father and single-mother families. Women tended to prefer healthy food such as fruits and dairy products, while men tended to prefer animal-source food such as meat, poultry, and aquatic products. We confirmed the heterogeneous preference between men and women by comparing the food consumption in single-mother and single-father children (Appendix A). We found that the consumption of cereals, vegetables, meat and poultry, and aquatic products declined after the mother separated from her husband while their consumption of fruits increased significantly. Accordingly, the intake of protein, calorie, carbohydrate, vitamin B, iron, and zinc of single-mother decreased. Differently, single-father children had a higher consumption of all food items except dairy products, particularly the consumption of meat and poultry and fruits. Their intake of most nutrients also increased after the mother left.

Different food preferences of parents resulted in different food compensations provided to single-parent children, which further caused different impacts on their nutrition intake. The insignificant impact on nutrition intake of single-mother children could be attributed to the fact that increasing consumption of eggs and fruits was offset by the decreasing consumption of meat and poultry, dairy products, vegetables, and cereals. The increasing intake of fat and of vitamins B and C of single-father children might be caused by the increasing consumption of meat, poultry, aquatic products, and vegetables.

After matching single-parent children living in urban, rich families and rural, poor families with their counterparts living in dual-parent families in urban and rural area, we found a stronger compensation in urban, rich families, which could be attributed to the higher food availability and lower cost of access to high-quality foods such as aquatic products and dairy products [23]. We also found that losing one parent had a strong negative impact on children’s diet and nutrition status in rural rich families, which could be attributed to the significant decrease in family income (88.04%). However, this conclusion should be explained with caution as we only have 18 observations from rural, rich families.

Several limitations in our study should be mentioned. First, we only kept the nearest two periods for each child due to data limitation so that there would be only a short-term impact estimated in our study and so that the long-term impact of one-parent absence on single-parent children remained for future research with long run panel survey data. Second, having one parent absent might also result in psychological harm to children, which might lead to apositia (lack of appetite) or surfeit [7,10]. These effects were difficult to identify in the present analysis, which might create bias in our estimation of the compensation effect. Third, the sample for single-parent children was small, so it was impossible to further estimate the impact on single-parent boys and single-parent girls separately. A longitudinal survey focusing on single-parent children would be needed. Fourth, single-parent children living with biological parents might be affected differently from those who lived with stepparents and foster parents. This issue also remains for future research with more specific data. Finally, positive food consumption does not necessarily lead to better health outcomes such as overweight and obesity risk, as health outcome depends not only on dietary pattern but also on many other factors such as physical activity and genetic factors.

## 5. Conclusions

Along with the changing attitudes and perception toward marriage and family values, it is expected that more single-parent families will arise in China in the future, which might lead to a soaring number of children growing up in one-parent families. Different from previous studies, our research contributed to the literature threefold. First, we found that the poor dietary status of single-parent children compared with dual-parent children was mainly caused by pretreatment heterogeneity. Single parents would compensate their children by providing more food. Second, we proposed a framework to decompose the total impact caused by the absence of one parent into an income effect and a compensation effect, which contributed to a deeper understanding of the mechanism on the association between changing family structures and children’ diets. Third, by comparing the results between single-mother and single-father children, we found that single mothers and single fathers provided different food compensation, which depended on their own preference. Quite possibly, the food compensation provided by single parents might not be good for children’ growth and health in light of the generally poor dietary knowledge of Chinese citizens [37]. In particular, if single parents give more money to their children as compensation, the children might tend to buy more processed food such as deep-fried and salty products, which could harm their growth and health in the future. We thus strongly recommended government to provide more dietary knowledge to parents, particularly single parents, which might be more effective than programs directly targeting single-parent children.

## Figures and Tables

**Table 1 nutrients-11-03077-t001:** Descriptive analysis of variables.

Variables	Mean (Standard Deviation)				ANOVA	
	Single Parent	Single Mother	Single Father	Dual Parent	Single/Dual	Three Groups
Income	19,153	18,259	21,171	27,462	17.82 ***	9.18 ***
	(18,824)	(18,730)	(18,990)	(32,143)	(0.00)	(0.00)
ln(income)	9.32	9.21	9.58	9.65	11.44 ***	7.48 ***
	(1.46)	(1.62)	(1.00)	(1.53)	(0.00)	(0.00)
Household size	5.66	5.35	6.36	4.68	71.01 ***	45.19 ***
	(2.25)	(2.13)	(2.37)	(1.74)	(0.00)	(0.00)
Children ratio	0.37	0.37	0.37	0.35	5.04 **	2.52 *
	(0.14)	(0.13)	(0.15)	(0.13)	(0.02)	(0.08)
Children age	8.84	8.66	9.22	9.76	13.00 ***	7.07 ***
	(3.75)	(3.76)	(3.72)	(4.04)	(0.00)	(0.00)
Children gender	0.49	0.51	0.44	0.46	0.82	0.89
	(0.50)	(0.50)	(0.50)	(0.50)	(0.37)	(0.41)
Household head age	47.05	45.78	49.90	45.78	2.11	3.78 **
	(15.58)	(14.72)	(17.10)	(13.31)	(0.15)	(0.02)
Household head activity	3.05	3.01	3.13	2.82	8.97 ***	4.77 ***
	(1.10)	(1.12)	(1.05)	(1.20)	(0.00)	(0.01)
Household head gender	0.49	0.58	0.29	0.35	20.32 ***	21.31 ***
	(0.50)	(0.49)	(0.46)	(0.48)	(0.00)	(0.00)
Household head education	1.32	1.35	1.26	1.77	31.96 ***	16.13 ***
	(1.00)	(0.93)	(1.14)	(1.28)	(0.00)	(0.00)
Urban dummy	0.88	0.87	0.91	0.67	51.29 ***	25.88 ***
	(0.33)	(0.34)	(0.29)	(0.47)	(0.00)	(0.00)
Time gap	2.79	2.84	2.67	2.75	0.21	0.60
	(1.36)	(1.40)	(1.26)	(1.23)	(0.65)	(0.55)
Observations	280	194	86	1948		

*, **, *** Statistically significant at 10%, 5%, and 1%, respectively. Values in brackets in the left four columns are standard deviations, and that in the right two columns are *p* values of ANOVA tests. Activity refers to the physical activity levels of the household head, which is measured according to occupation type and ranged from 1 to 6, namely, 1 = no physical activity; 2 = very light physical activity, working in a sitting position (for example, office worker, or watch repairer); 3 = light physical activity, working in a standing position (for example, sales person or teacher); 4 = moderate physical activity (for example, student or driver); 5 = heavy physical activity (for example, farmer or dancer); and 6 = very heavy physical activity (for example, loader, logger, or miner). Education measures the education level of the household head, which ranges from 0 (no education) to 6 (PhD degree).

**Table 2 nutrients-11-03077-t002:** Comparison of food consumption and nutrition intake between single-parent children and dual-parent children before matching.

Category	Children	Dual-Parent	Single-Parent		Single-Mother		Single-Father		Three Groups
	Dietary Indicators	Mean	Mean	ANOVA	Mean	ANOVA	Mean	ANOVA	ANOVA
Food consumption	Cereals (g)	288.70	282.23	0.44	297.42	0.56	247.98	5.92 **	3.34 **
		(153.34)	(150.73)	(0.51)	(162.11)	(0.45)	(114.85)	(0.02)	(0.04)
	Vegetables (g)	189.47	192.75	0.16	195.27	0.35	187.07	0.03	0.20
		(128.78)	(139.13)	(0.69)	(144.05)	(0.55)	(127.94)	(0.87)	(0.82)
	Meat and poultry (g)	71.43	58.87	8.67 ***	63.15	2.67 *	49.23	9.12 ***	5.63 ***
		(67.40)	(61.83)	(0.00)	(66.60)	(0.10)	(48.37)	(0.00)	(0.00)
	Aquatic products (g)	18.71	8.74	17.90 ***	9.99	9.58 ***	5.91	9.39 ***	9.31 ***
		(38.50)	(22.42)	(0.00)	(23.66)	(0.00)	(19.15)	(0.00)	(0.00)
	Eggs (g)	23.76	19.40	5.56 **	20.28	2.53	17.43	3.91 **	3.07 **
		(29.20)	(26.74)	(0.02)	(27.39)	(0.11)	(25.24)	(0.05)	(0.05)
	Dairy products (g)	14.82	7.45	4.44 **	6.80	3.73 **	8.92	0.91	2.26 *
		(56.47)	(40.10)	(0.04)	(38.57)	(0.05)	(43.55)	(0.34)	(0.10)
	Fruits (g)	59.63	43.34	5.53 **	46.58	2.51	36.04	3.88 **	3.04 **
		(109.86)	(98.08)	(0.02)	(105.82)	(0.11)	(77.91)	(0.05)	(0.05)
Nutrition intake	Calorie (kilocalorie)	1212.59	1176.95	1.10	1226.49	0.12	1065.18	6.42 ***	3.31 **
		(529.36)	(542.71)	(0.29)	(557.16)	(0.73)	(493.63)	(0.01)	(0.04)
	Carbohydrate (g)	205.01	209.06	0.41	217.97	3.00 *	188.97	2.21	2.76*
		(98.23)	(104.82)	(0.52)	(109.97)	(0.08)	(89.52)	(0.14)	(0.06)
	Fat (g)	26.86	22.71	12.36 ***	23.43	6.05 ***	21.08	7.96 ***	6.66 ***
		(18.67)	(17.18)	(0.00)	(17.24)	(0.01)	(17.05)	(0.00)	(0.00)
	Protein (g)	44.20	40.25	8.82 ***	42.46	1.22	35.26	14.98 ***	7.99 ***
		(21.15)	(18.48)	(0.00)	(18.93)	(0.27)	(16.46)	(0.00)	(0.00)
	Vitamin A (Retinol Equivalent, μg)	302.62	296.08	0.06	317.82	0.23	247.04	1.43	0.88
		(427.63)	(346.13)	(0.81)	(380.33)	(0.63)	(247.59)	(0.23)	(0.41)
	Vitamin B (mg)	10.75	10.05	3.85 **	10.44	0.56	9.19	6.49 ***	3.43 **
		(5.61)	(5.28)	(0.05)	(5.54)	(0.45)	(4.56)	(0.01)	(0.03)
	Vitamin C (mg)	49.61	48.59	0.18	48.49	0.16	48.82	0.04	0.09
		(36.70)	(44.48)	(0.67)	(42.12)	(0.69)	(49.67)	(0.85)	(0.91)
	Vitamin E (mg)	7.61	6.67	6.44 ***	7.00	1.90	5.90	6.87 ***	4.29 ***
		(5.93)	(4.97)	(0.01)	(4.78)	(0.17)	(5.34)	(0.00)	(0.01)
	Calcium (mg)	247.74	227.52	2.86 *	229.57	1.68	222.90	1.42	1.47
		(187.92)	(181.01)	(0.09)	(167.23)	(0.19)	(209.79)	(0.23)	(0.23)
	Iron (mg)	13.08	12.61	0.82	13.34	0.19	10.96	5.67 **	3.01 **
		(8.14)	(7.39)	(0.37)	(7.98)	(0.67)	(5.57)	(0.02)	(0.05)
	Zinc (mg)	7.12	6.85	1.66	7.08	0.02	6.32	4.86 **	2.42 *
		(3.30)	(3.27)	(0.20)	(3.37)	(0.88)	(3.01)	(0.03)	(0.09)
	Selenium (μg)	29.86	25.53	12.63 ***	27.69	2.24	20.67	19.03 ***	10.39 ***
		(19.41)	(16.34)	(0.00)	(17.96)	(0.13)	(10.46)	(0.00)	(0.00)
Number of Observations		1948	280		194		86		

The differences refer to difference between dual-parent children and all types of single-parent children. Values in brackets under the mean values are standard deviations, and that for ANOVA are *p* values of ANOVA tests. *, **, *** Statistically significant at 10%, 5%, and 1%, respectively.

**Table 3 nutrients-11-03077-t003:** Average treatment effect of the treated (ATT) of single-parent children for various dietary indicators.

ATT		Single-Parent	Single-Mother	Single-Father
Food consumption	Cereals (g)	−5.30	−1.99	−28.40 **
		(13.42)	(14.09)	(13.98)
	Vegetables (g)	7.54	−4.27	15.65
		(11.56)	(12.17)	(12.30)
	Meat and poultry (g)	3.65	−0.27	11.73 **
		(5.38)	(5.80)	(5.45)
	Aquatic products (g)	1.24	0.08	4.81 **
		(2.45)	(2.68)	(2.44)
	Eggs (g)	5.02 **	6.88 ***	2.04
		(2.36)	(2.50)	(2.51)
	Dairy products (g)	−1.97	−3.44	−4.72
		(3.79)	(4.15)	(3.98)
	Fruits (g)	13.54	17.36 *	3.18
		(8.35)	(8.99)	(8.84)
Nutrition intake	Calorie (kilocalorie)	49.05	35.11	−19.78
		(45.96)	(48.19)	(50.85)
	Carbohydrate (g)	7.85	8.87	−13.51
		(8.84)	(9.29)	(9.83)
	Fat (g)	2.17	0.18	4.32 ***
		(1.50)	(1.56)	(1.65)
	Protein (g)	−0.50	−0.88	−1.52
		(1.66)	(1.76)	(1.78)
	Vitamin A (Retinol Equivalent, μg)	−19.89	−8.04	−33.45
		(31.11)	(33.72)	(30.11)
	Vitamin B (mg)	0.10	−0.46	0.88 *
		(0.46)	(0.49)	(0.46)
	Vitamin C (mg)	5.00	1.24	11.93 ***
		(3.52)	(3.48)	(4.35)
	Vitamin E (mg)	0.16	0.07	−0.08
		(0.45)	(0.46)	(0.53)
	Calcium (mg)	5.11	−5.89	26.70
		(15.46)	(15.73)	(18.70)
	Iron (mg)	−0.29	−0.56	−0.46
		(0.66)	(0.72)	(0.65)
	Zinc (mg)	0.16	0.06	−0.14
		(0.28)	(0.29)	(0.31)
	Selenium (μg)	1.73	2.10	−1.67
		(1.52)	(1.60)	(1.77)
Logit model statistics	LR chi^2^(19)	76.18	50.24	73.19
	*p*	0.00	0.00	0.00
	R^2^	0.09	0.08	0.21
	Observations	1114	1071	1017

Values represent an average treatment effect of treated (ATT) after one parent leaves. Values in brackets are standard deviation. *, **, *** Statistically significant at 10%, 5%, and 1%, respectively. LR Chi2(19) refers to the likelihood ratio test of the significance of the model.

**Table 4 nutrients-11-03077-t004:** Income effect and compensation effect.

Children		Single-Parent			Single-Mother			Single-Father		
		ATT	Income Effect	Compensation Effect	ATT	Income Effect	Compensation Effect	ATT	Income Effect	Compensation Effect
Food consumption	Cereals (g)	−5.30	1.84	−7.14	−1.99	1.68	−3.67	−28.40	2.00	−30.40
	Vegetables (g)	7.54	−0.15	7.47	−4.27	−0.75	−4.88	15.65	4.68	14.73
	Meat and poultry (g)	3.65	−2.50	2.54	−0.27	−2.52	−2.32	11.73	−0.94	11.92
	Aquatic products (g)	1.24	0.42	1.43	0.08	0.31	0.33	4.81	1.07	4.60
	Eggs (g)	5.02	0.16	5.09	6.88	0.11	6.97	2.04	0.42	1.96
	Dairy products (g)	−1.97	0.41	−1.79	−3.44	0.46	−3.06	−4.72	−0.21	−4.68
	Fruits (g)	13.54	0.86	13.92	17.36	−0.22	17.18	3.18	8.96	1.42
Nutrition intake	Calorie (kilocalorie)	49.05	−0.38	48.88	35.11	−1.95	33.52	−19.78	12.30	−22.20
	Carbohydrate (g)	7.85	1.51	8.52	8.87	1.44	10.04	−13.51	1.19	−13.75
	Fat (g)	2.17	−0.66	1.88	0.18	−0.75	−0.44	4.32	0.47	4.23
	Protein (g)	−0.50	0.09	−0.46	−0.88	−0.04	−0.91	−1.52	1.07	−1.73
	Vitamin A (μg)	−19.89	−1.27	−20.45	−8.04	−2.21	−9.84	−33.45	1.14	−33.67
	Vitamin B (mg)	0.10	−0.03	0.09	−0.46	−0.04	−0.49	0.88	0.11	0.86
	Vitamin C (mg)	5.00	0.15	5.07	1.24	0.07	1.30	11.93	0.68	11.80
	Vitamin E (mg)	0.16	0.17	0.23	0.07	0.11	0.16	−0.08	0.52	−0.18
	Calcium (mg)	5.11	1.01	5.56	−5.89	−1.59	−7.18	26.70	21.06	22.56
	Iron (mg)	−0.29	0.01	−0.29	−0.56	−0.01	−0.57	−0.46	0.14	−0.49
	Zinc (mg)	0.16	−0.04	0.14	0.06	−0.06	0.01	−0.14	0.13	−0.17
	Selenium (μg)	1.73	−0.20	1.64	2.10	−0.28	1.87	−1.67	0.56	−1.78

Vitamin A is measured by retinol equivalent.

**Table 5 nutrients-11-03077-t005:** ATT of single-parent children in rural, poor families and urban, rich families.

Children		Urban, Rich Family			Rural, Poor Family		
		ATT	Income Effect	Compensation Effect	ATT	Income Effect	Compensation Effect
Food consumption	Cereals (g)	11.64	14.50	−2.86	133.17 **	−10.49	143.66
	Vegetables (g)	38.71 **	10.74	27.97	33.30	−11.00	44.30
	Meat and poultry (g)	8.83	−1.58	10.41	−13.94	10.93	−24.87
	Aquatic products (g)	2.58	−0.22	2.80	−0.31	−1.26	0.95
	Eggs (g)	−0.59	0.00	−0.59	0.53	−5.43	5.96
	Dairy products (g)	3.57	1.38	2.19	−19.51	1.63	−21.14
	Fruits (g)	25.59 **	12.53	13.06	30.97	−5.06	36.03
Nutrition intake	Calorie (kilocalorie)	191.63 ***	48.22	143.41	159.74	−56.63	216.37
	Carbohydrate (g)	31.07 ***	8.60	22.47	49.78	−16.83	66.61
	Fat (g)	7.17 ***	1.30	5.87	−6.19	1.10	−7.29
	Protein (g)	3.29	0.85	2.44	5.26	−1.32	6.58
	Vitamin A (μg)	−105.06 **	−8.13	−96.93	247.55 **	−29.74	277.29
	Vitamin B (mg)	1.38 **	0.01	1.37	1.07	−0.59	1.66
	Vitamin C (mg)	10.32 **	−0.37	10.69	19.61 *	4.34	15.27
	Vitamin E (mg)	2.02 ***	−0.14	2.16	0.00	−0.09	0.09
	Calcium (mg)	49.66 **	−3.15	52.81	65.72	28.09	37.63
	Iron (mg)	1.71 *	−0.12	1.83	1.77	−0.13	1.90
	Zinc (mg)	0.66 *	0.03	0.63	0.66	−0.38	1.04
	Selenium (μg)	5.21 **	1.53	3.68	−0.37	−2.27	1.90

Vitamin A is measured by retinol equivalent. ATT refers to average treatment effect of treated after losing one parent. *, **, *** Statistically significant at 10%, 5%, and 1%, respectively. The average income change in urban, rich families and rural, poor families between two time periods are −18,385 (−58.93%) and 11,328 (183.33%) respectively.

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
