# Peer review of "The Impact of Having One Parent Absent on Children’ Food Consumption and Nutrition in China"

_nutrients, 2019, doi:10.3390/nu11123077_

Round 1

Reviewer 1 Report

This paper presents a methodologically sophisticated exploration of the differences in food consumption between single- and dual-parent families' children in China.  The use of propensity score analysis enabled findings to be revealed that were more accurate than what would have been obscured by sample heterogeneity.

My main concern with the study was the difference in sample sizes between the dual- and single-parent groups.  I wondered how the difference in statistical power between the groups affected the outcome of the results.  There are ANOVA tests, such as Welch's ANOVA, that account for unequal sample sizes that could be run with two groups, or single-mother, single-father, and dual-parent groups could be compared simultaneously with this method to account for the sample size issue.

I also thought that "urban rich" vs. "rural poor" conflated location with SES, so it was hard to determine which variable was responsible for obtained effects.  It would be helpful to test urban poor and rural rich, as well, to answer this question.

I thought that the literature review needed more information on food consumption patterns of single- vs. dual-parent families in China, as the information provided was limited.  If none is available, more general information regarding eating patterns in China is warranted.

Author Response

Review report 1:

This paper presents a methodologically sophisticated exploration of the differences in food consumption between single- and dual-parent families' children in China.  The use of propensity score analysis enabled findings to be revealed that were more accurate than what would have been obscured by sample heterogeneity.

Responses: Thank you very much for your positive evaluation of our research and valuable comments. We have revised the manucript according to the comments proposed by you and another reviewer. The point-to-point responses are as follows:

My main concern with the study was the difference in sample sizes between the dual- and single-parent groups.  I wondered how the difference in statistical power between the groups affected the outcome of the results.  There are ANOVA tests, such as Welch's ANOVA, that account for unequal sample sizes that could be run with two groups, or single-mother, single-father, and dual-parent groups could be compared simultaneously with this method to account for the sample size issue.

Responses: Thank you for your suggestion. We have adopted the ANOVA tests to retest the equality of mean values between single-parent children and dual-parent children, which accounts for unequal sample sizes between different groups. The mean equality between single-parent children and dual-parent children is conducted using two-group ANOVA test, and the mean equality between dual-parent children, single-father children, and single-mother children is conducted using three-group ANOVA test. Results are presented in Table 1&2.

I also thought that "urban rich" vs. "rural poor" conflated location with SES, so it was hard to determine which variable was responsible for obtained effects.  It would be helpful to test urban poor and rural rich, as well, to answer this question.

Responses: Thank you for your suggestion. The reason to compare urban rich and rural poor is to investigate whether heterogeneous families compensate single-parent children differently. Because urban rich and rural poor families are very different in terms of SES and food availability, the difference of compensation should be more apparent. Our motivation is to see whether the compensation differs in different families, rather than looking for the determinants of these difference. So we still mainly focus on the comparison between urban rich and rural poor, but we also present the comparison between urban poor and rural rich in the supplementary file (S table 10). We still find positive compensation in urban poor families, but not as strong as that in urban rich families. To the contrary, we find negative ATT and compensation in rural poor families for several food and nutrition intake.

I thought that the literature review needed more information on food consumption patterns of single- vs. dual-parent families in China, as the information provided was limited.  If none is available, more general information regarding eating patterns in China is warranted.

Responses: This is a good comment. Yes, we did not find too many paper working on food consumption patterns of single-parent and dual-parent families in China. But we did find a lot of literature investigating the transition of eating patterns in China. We have added some information about the nutrition transition in the introduction section as follows:

“Moreover, previous literature already showed that China was undergoing a rapid nutrition transition in the past decades. The traditional Chinese diet, which is high in complex carbohydrates and fiber, had been gradually replaced by a refined food and Western food diet, which is high in fat, saturated fat, and sugar [11] [21, 22] [23]. As a result, the food accessibility and dietary diversity had been significantly improved in China.”

Reviewer 2 Report

In this study, the authors seeked to indirectly assess nutritional status of single-parent children compared to dual-parent children in Chinese families. They compared consumption of 7 representative food items and intake of 12 nutrients. The authors gathered data collected by the China Health and Nutrition survey. They report that the absence of a parent from the family home does not negatively affect food consumption and nutrient intake in children below the age of 18. 

I'd suggest specifying in the title that this study was conducted using data from a Chinese population.

The authors report that absence of a parent results in compensation. I'd suggest discussing how compensating for the absence of a parent with food can have long-term detrimental effects on a child as it can lead to obesity and/or other food related disorders. 

Author Response

Review report 2:

In this study, the authors seeked to indirectly assess nutritional status of single-parent children compared to dual-parent children in Chinese families. They compared consumption of 7 representative food items and intake of 12 nutrients. The authors gathered data collected by the China Health and Nutrition survey. They report that the absence of a parent from the family home does not negatively affect food consumption and nutrient intake in children below the age of 18. 

Responses: Thank you for your positive evaluation of our paper. We have revised the manuscript according to the comments proposed by you and another reviewer. The point-to-point responses are as follows:

I'd suggest specifying in the title that this study was conducted using data from a Chinese population.

Responses: Thank you very much for your suggestion. We have changed the title to “The Impact of Having One Parent Absent on Children’ Food Consumption and Nutrition in China”

The authors report that absence of a parent results in compensation. I'd suggest discussing how compensating for the absence of a parent with food can have long-term detrimental effects on a child as it can lead to obesity and/or other food related disorders. 

Responses: This is a very good suggestion. In fact we are working on a new paper which focuses on comparing the health status such as obesity/overweight risk between single-parent and dual-parent children. We do not present the health outcome in this paper because we loss too many observations after matching food consumption data with physical examination data. Below you can find the preliminary results of standardized BMI, overweight and obesity risk using merged data. We can find that losing mother will have a negative impact on children’s BMI. So that higher food compensation does not necessarily lead to better nutrition and weight gains. However, this result should be explained with caution as we only have very limit observations. We will work on that in the future.

Pairs

Health

indicator

Initial period

Following up period

DID

No. of observations

Dual-

Single-

Diff.

Dual-

Single-

Diff.

Single-parent v.s. Dual-parent

BMI_SD

1.002

0.969

-0.033***

0.976

0.950

-0.026**

0.007

102

Overweight

0.207

0.195

-0.012

0.188

0.146

-0.042

-0.030

Obesity

0.145

0.154

0.010

0.118

0.081

-0.037*

-0.046

Single-mother v.s. Dual-parent

BMI_SD

1.005

0.968

-0.037***

0.982

0.963

-0.019*

0.018

69

Overweight

0.219

0.153

0.066**

0.200

0.141

-0.059**

0.008

Obesity

0.154

0.118

-0.037*

0.126

0.071

-0.055**

-0.018

Single-father v.s. Dual-parent

BMI_SD

0.986

0.985

0.000

0.959

0.927

-0.033**

-0.032*

30

Overweight

0.183

0.306

0.123***

0.148

0.139

-0.009

-0.132***

Obesity

0.128

0.250

0.122***

0.084

0.083

0.000

-0.122***
